# An evidence-based 3D reconstruction of *Asteroxylon mackiei*, the most complex plant preserved from the Rhynie chert

Alexander J Hetherington[1†]*, Siobhán L Bridson[1], Anna Lee Jones[1,2‡],
Hagen Hass[3], Hans Kerp[3], Liam Dolan[1§]*

[1]Department of Plant Sciences, University of Oxford, Oxford, United Kingdom;
[2]Department of Plant Sciences, University of Cambridge, Cambridge, United
Kingdom; [3]Research Group for Palaeobotany, Institute for Geology and
Palaeontology, Westfälische Wilhelms-Universität Münster, Münster, Germany

*For correspondence:
sandy.hetherington@ed.ac.uk
(AJH);
liam.dolan@gmi.oeaw.ac.at (LD)

Present address: † Institute of
Molecular Plant Sciences, School
of Biological Sciences, University
of Edinburgh, Edinburgh, United
Kingdom; ‡ Oxford Long-term
Ecology Laboratory, Department
of Zoology, University of Oxford,
Oxford, United Kingdom; §
Gregor Mendel Institute of
Molecular Plant Biology GmbH,
Vienna, Austria

Competing interests: The
authors declare that no
competing interests exist.

Reviewing editor: Min Zhu,
Chinese Academy of Sciences,
China

**Abstract** The Early Devonian Rhynie chert preserves the earliest terrestrial ecosystem and informs our understanding of early life on land. However, our knowledge of the 3D structure, and development of these plants is still rudimentary. Here we used digital 3D reconstruction techniques to produce the first well-evidenced reconstruction of the structure and development of the rooting system of the lycopsid *Asteroxylon mackiei*, the most complex plant in the Rhynie chert. The reconstruction reveals the organisation of the three distinct axis types – leafy shoot axes, root-bearing axes, and rooting axes – in the body plan. Combining this reconstruction with developmental data from fossilised meristems, we demonstrate that the *A. mackiei* rooting axis – a transitional lycophyte organ between the rootless ancestral state and true roots – developed from root-bearing axes by anisotomous dichotomy. Our discovery demonstrates how this unique organ developed and highlights the value of evidence-based reconstructions for understanding the development and evolution of the first complex vascular plants on Earth.

## Introduction

The Silurian–Devonian terrestrial revolution saw the evolution of vascular plants with complex bodies comprising distinct roots, root-bearing organs, shoots, and leaves from morphologically simpler ancestors characterised by networks of undifferentiated axes (*Bateman et al., 1998*; *Gensel and Edwards, 2001*; *Kenrick and Crane, 1997*; *Stewart and Rothwell, 1993*; *Xue et al., 2018*). The 407-million-year-old, Pragian–?earliest Emsian, Early Devonian (*Wellman, 2006*) Rhynie chert fossil site provides a unique insight into the structure of plants during this key time in plant evolution. The Rhynie chert preserves an entire Early Devonian hot spring ecosystem, where plants, animals, fungi, and microbes are preserved in situ (*Edwards et al., 2018*; *Garwood et al., 2020*; *Rice et al., 2002*; *Strullu-Derrien et al., 2019*; *Wellman, 2018*). The exceptional preservation has been crucial for our understanding of early land plant evolution because it is the earliest time point in the fossil record where cellular details of rhizoid-based rooting systems, germinating spores, and fossilised meristems are preserved (*Edwards, 2003*; *Hetherington and Dolan, 2018a*; *Hetherington and Dolan, 2018b*; *Kerp, 2018*; *Lyon, 1957*; *Taylor et al., 2005*). Most of the detailed cellular information about these organisms comes from sectioned material. While the cellular detail that can be observed in these sections allows high-resolution reconstruction of tissue systems, the three-dimensional relationship between the cells, tissue, and organs is obscured. This makes generating accurate reconstructions of body plans difficult (*Edwards, 2003*; *Kidston and Lang, 1921*). Furthermore, reconstructions that have been published are based on combining material from thin sections from multiple individuals (*Kidston and Lang, 1921*). These sampling problems mean that key features of the body plans of

these organisms are missing in reconstructions. This is particularly problematic for larger, more complex species in the Rhynie chert, such as the lycopsid *Asteroxylon mackiei* (*Bhutta, 1969*; *Edwards, 2003*; *Edwards et al., 2018*; *Hetherington and Dolan, 2018a*; *Kerp, 2018*; *Kerp et al., 2013*; *Kidston and Lang, 1920*; *Kidston and Lang, 1921*).

*A. mackiei* has been reconstructed as a plant that is approximately 30 cm high (*Bhutta, 1969*; *Edwards, 2003*), with highly branched shoot and rooting systems (*Chaloner and MacDonald, 1980*; *Kerp, 2018*; *Kerp et al., 2013*; *Kidston and Lang, 1920*; *Kidston and Lang, 1921*). It holds an important phylogenetic position for understanding root and leaf evolution in lycophytes because it is a member of the earliest diverging lineage of the lycopsids, the Drepanophycales (*Kenrick and Crane, 1997*), and both the rooting axes and leaves of *A. mackiei* developed some but not all defining characteristics of roots and leaves in more derived species (*Bower, 1908*; *Hetherington and Dolan, 2018a*; *Kenrick, 2002*; *Kidston and Lang, 1920*). However, the precise number of distinct axis types and their interconnection is still unclear (*Bhutta, 1969*; *Kidston and Lang, 1920*; *Kidston and Lang, 1921*). Without a complete understanding of the growth habit of *A. mackiei*, it is not possible to compare its structure with living lycopsids or other drepanophycalean lycopsids found in Devonian compression fossils. The drepanophycalean lycopsids are the earliest group of land plants in the fossil record with complex body plans comprising distinct rooting axes, root-bearing organs, and leafy shoots (*Gensel et al., 2001*; *Gensel and Edwards, 2001*; *Hueber, 1992*; *Kenrick and Crane, 1997*; *Lang and Cookson, 1935*; *Matsunaga and Tomescu, 2016*; *Matsunaga and Tomescu, 2017*; *Stewart and Rothwell, 1993*). Their evolution, radiation, and spread across all continents contributed to the transformation of the terrestrial environment through their impact on soil formation and stabilisation, surface hydrology, and silicate weathering (*Algeo and Scheckler, 1998*; *Gibling and Davies, 2012*; *Matsunaga and Tomescu, 2016*; *Xue et al., 2016*). Given the recognition of the importance of the drepanophycalean lycopsids in the evolution of complex body plans and changes to global nutrient and hydrological cycles, we generated a reconstruction of the 3D structure of *A. mackiei* based entirely on serial sections from complete specimens fossilised in situ.

Here we report the 3D reconstruction of *A. mackiei* based on both morphology and anatomy of two different plants. This reconstruction allowed us to define the 3D organisation of the three axis types of the *A. mackiei* body and to describe how the rooting system developed.

## Results

To discover the structure and infer the development of the lycopsid *A. mackiei*, we produced a series of 31 consecutive thick sections through a block of Rhynie chert that preserved a branched network of connected *A. mackiei* axes in situ (*Figure 1—figure supplement 1*, *Figure 1—figure supplement 2*). Using images of these thick sections, we digitally reconstructed the *A. mackiei* plant in a volume of 4.8 cm in length, 3.5 cm in width, and 2.8 cm in height (*Figure 1A,B,G Video 1*), which, to our knowledge, represents the largest evidence-based reconstruction for any Rhynie chert plant to date. We distinguished three distinct axis types in a single individual plant that we designate: leafy shoot axes, root-bearing axes, and rooting axes.

### Leafy shoot axes

The majority of the axes in our reconstruction were leafy shoots (*Figure 1A,B*, *Video 1*). Leafy shoot axes developed leaves, abundant stomata, and a characteristic internal anatomy including a stellate xylem, many leaf traces, and trabecular zone as reported for *A. mackiei* (*Bhutta, 1969*; *Kerp, 2018*; *Kerp et al., 2013*; *Kidston and Lang, 1920*; *Kidston and Lang, 1921*; *Lyon, 1964*; *Figure 1C,D*). The presence of a geopetally infilled void in the sections allowed us to determine the orientation of axes relative to the gravitational vector because the silica that partly fills the void space acts as a spirit level indicating the direction of the gravity vector when it was deposited (*Trewin and Fayers, 2015*). The main axis present in each of the thick sections was horizontal (plagiotropic) (*Figure 1—figure supplement 1*). Four leafy shoot axes with similar anatomy attached to the main axis at anisotomous branch points; an anisotomous branch point is a description of morphology and means that the diameters of the two axes connected at a branch point are different (*Gola, 2014*; *Imaichi, 2008*; *Ollgaard, 1979*; *Yin and Meicenheimer, 2017*). The diameter of the main plagiotropic leafy shoot was ca. 1 cm and the thinner leafy shoots attached at branch points were ca. 0.6 cm.

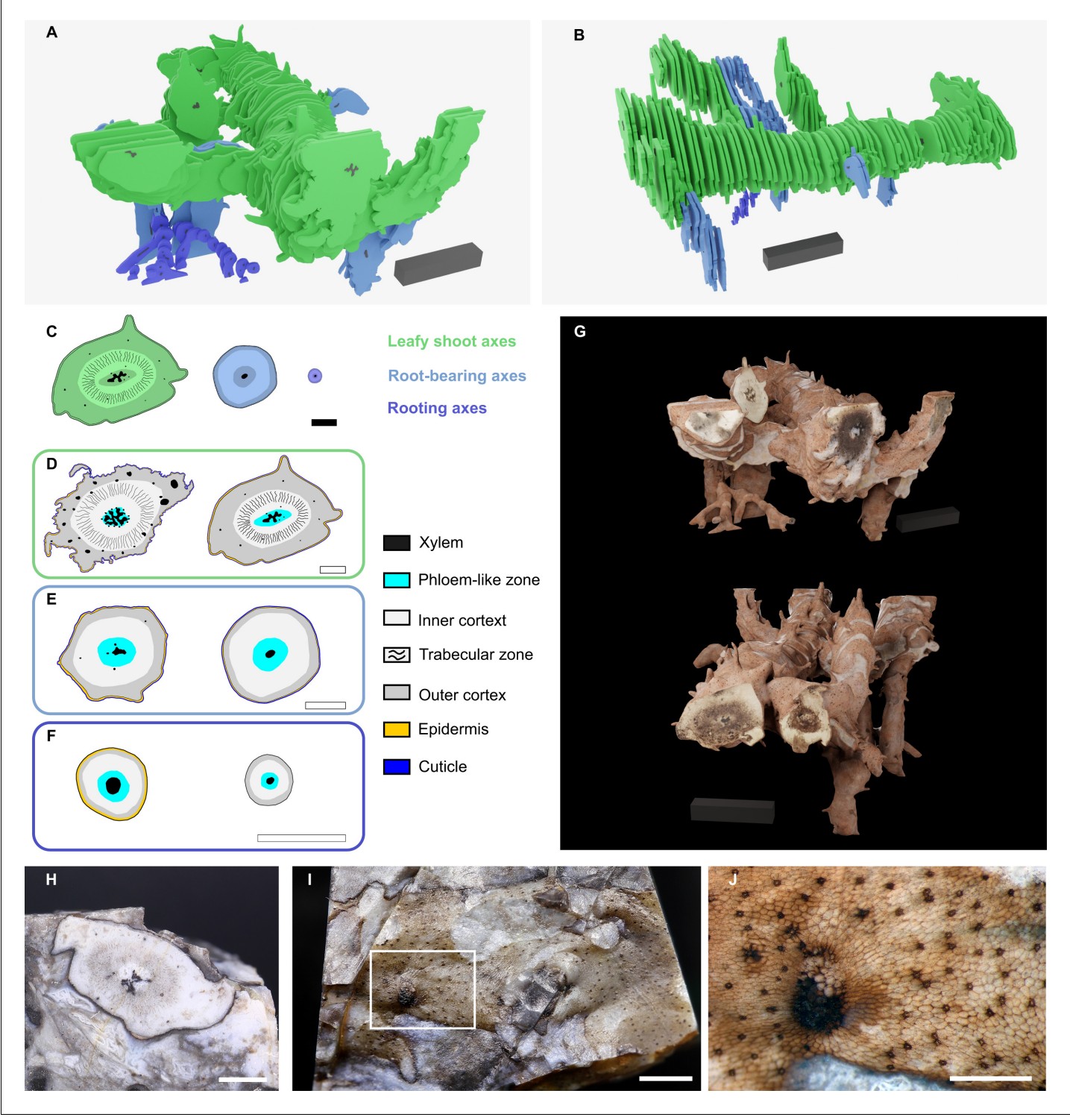

**Figure 1.** The body plan of *Asteroxylon mackiei* was composed of three distinct axes: leafy shoot axes, root-bearing axes, and rooting axes. (A, B) 3D reconstruction of *A. mackiei* based on a series of 31 thick sections. (C) Representative examples of transverse sections through the three main axis types colour coded to match their colours in the 3D reconstruction (A, B), leafy shoot axes in green, root-bearing axes in blue, and rooting axes in purple. (D-F) Line drawings of representatives of each of the three main axis types illustrating their anatomy. Examples of two representative leafy shoots (D), root-bearing axes (E) and rooting axes (F). (G) An artist's impression of the complete fossil rooting system reconstructed from thick sections. (H-J) Example of a plagiotropic leafy shoot exposed on the surface of a block of chert Pb 2020_01. (H) End on view of the block of chert with *A. mackiei* leafy shoot axis cut in transverse section. (I) Same block as in (H) showing the surface of the axis with brown cuticle and sparse covering of leaves. (J) Higher magnification image of white box in (I) showing a single leaf base and abundant stomata. Line drawings of *A. mackiei* axes based on specimen

*Figure 1 continued on next page*

*Figure 1 continued*

accession codes: GLAHM Kid. 2479 and Pb 4181 (**D**), Bhutta Collection RCA 13 and RCA 113 (**E**), GLAHM Kid 2471 and GLAHM Kid 2477 (**F**). 3D scale bar 1 × 0.1 × 0.1 cm (**A, B, G**). Scale bars, 2 mm (**C–F, H, I**), 1 mm (**J**). (**G**) Illustrations by Matt Humpage (https://twitter.com/Matt_Humpage). The online version of this article includes the following figure supplement(s) for figure 1:

**Figure supplement 1.** Geopetally infilled void allowed growth orientation to be established.
**Figure supplement 2.** *A. mackiei* axes were preserved in original growth position.

Some of the thinner leafy shoots were orientated closer to the vertical, indicating orthotropic growth orientation (*Figure 1A,B,G*). Although our reconstruction did not include connections between these orthotropic axes and previously described fertile axes, it is likely that some of these orthotropic leafy shoot axes were connected to fertile axes (*Bhutta, 1969*; *Kerp et al., 2013*; *Lyon, 1964*). The most noticeable differences between the plagiotropic leafy shoots described here and orthotropic shoots described previously (*Bhutta, 1969*; *Kerp et al., 2013*; *Kidston and Lang, 1920*; *Kidston and Lang, 1921*; *Lyon, 1964*) are that the xylem was less lobed and there were fewer leaf traces in the plagiotropic leafy shoot axes than in the orthotropic leafy shoot axes (*Figure 1D*). Fewer leaf traces in plagiotropic regions is consistent with a lower leaf density on these axes than in orthotropic axes, a feature demonstrated in detail by the discovery of an isolated plagiotropic leafy shoot with sparse covering of leaves preserved on the exterior of a block of chert (*Figure 1H,J*). We conclude that *A. mackiei* developed plagiotropic and orthotropic leafy shoot axes with similar anatomy.

## Root-bearing axes

Root-bearing axes of *A. mackiei* were attached to leafy shoot axes at anisotomous branch points, where the thinner daughter axis developed as a root-bearing axis and the thicker daughter axis developed as a leafy shoot axis (*Figures 1* and *2*). Diameters of root-bearing axes were ca. 0.35 cm compared to leafy shoots axes typically over 0.6 cm. In one of the five examples (*Figure 2A–C*), the root-bearing axis was attached directly to the main leafy shoot axis. In the four other examples (*Figure 2D–I*), root-bearing axes were attached to side branches of the main leafy shoot. These branches were termed first-order leafy shoots because they were separated from the main shoot by a single branching event. Root-bearing axes attached to first-order leafy shoot axes close to where the latter attached to the main shoot. The branch arrangement where two adjacent anisotomous branches originate close to each other is termed k-branching (*Chomicki et al., 2017*; *Edwards, 1994*; *Gensel et al., 2001*; *Gensel and Berry, 2001*; *Gerrienne, 1988*; *Matsunaga and Tomescu, 2016*; *Matsunaga and Tomescu, 2017*). The root-bearing axes of the Drepanophycalean lycopsid *Sengelia radicans* (*Matsunaga and Tomescu, 2016*; *Matsunaga and Tomescu, 2017*) are attached to leafy shoot axes at k-branch points. In both *A. mackiei* and *S. radicans*, root-bearing axes developed an epidermis and cuticle with occasional stomata and scale leaves. In the root-bearing axes of *A. mackiei* where anatomy could be investigated the xylem strand was elliptical, not lobed as in leafy shoot axes, and there were few or no leaf traces, which distinguishes them from leafy shoots in which leaf traces were abundant (*Figure 1E*). Root-bearing axes were aligned with the gravity vector, indicating strong positive gravitropic growth (*Figures 1A,B* and *2*). These differences in anatomy and morphology between root-bearing axes and leafy shoots demonstrate that root-bearing axes were a distinct axis type and not merely a transitional zone between two axis types as previously suggested (*Bhutta, 1969*; *Kidston and Lang, 1920*; *Kidston and Lang,*

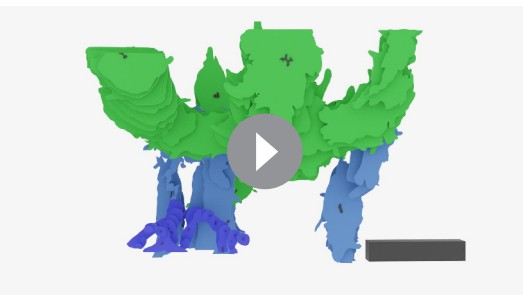

**Video 1.** 3D reconstruction of *A.mackiei* based on serial thick sections. A 3D reconstruction based on a series of 31 thick sections deposited in the collection of the Forschungsstelle für Geologie und Paläontologie, Westfälische Wilhelms-Universität, Münster, Germany under the accession numbers Pb 4161–4191. Leafy shoot axes in green, root-bearing axes in blue, and rooting axes in purple. 3D scale bar 1 × 0.1 × 0.1 cm. https://elifesciences.org/articles/69447#video1

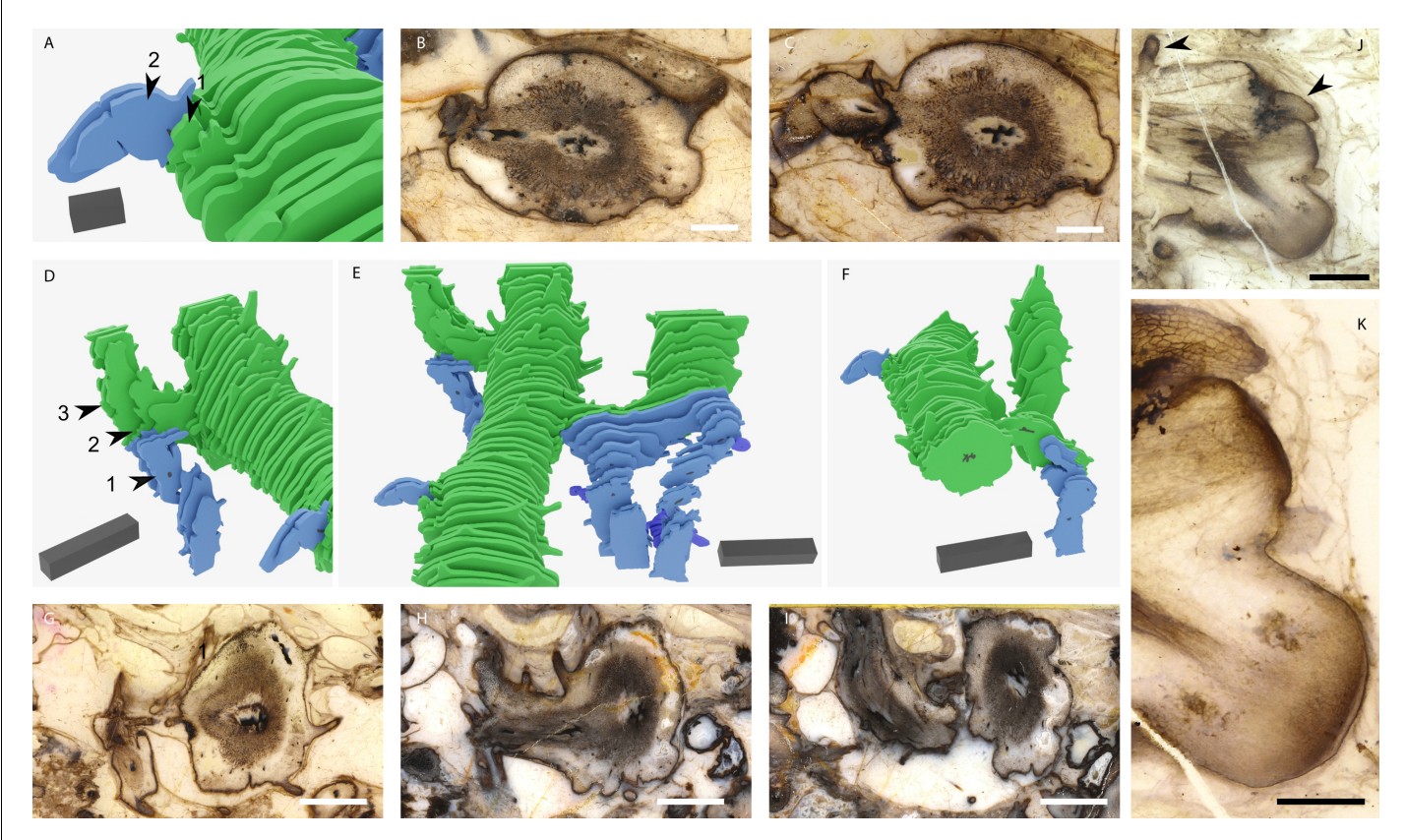

**Figure 2.** Root-bearing axes attached to leafy shoots at anisotomous branch points. Images showing the attachment of root-bearing axes shown in blue to leafy shoots shown in green based on our 3D reconstruction (**A, D–F**) and the thick sections used to create the reconstruction (**B, C, G–I**). (**A**) A root-bearing axis shown in blue attached to the side of the larger plagiotropic leafy shoot axis shown in green. (**B**) The thick section represented by arrow one in (**A**) showing a transverse section through the leafy shoot at the point of branching. The black xylem trace of the rooting axis is located to the left of the cross shaped xylem at the centre of the leafy shoot axis. (**C**) Thick section represented by arrow two in (**A**) showing the free root-bearing axis with small rounded xylem trace compared to the larger cross shaped xylem in the leafy shoot. (**D-F**) Examples based on the reconstruction of *A. mackiei* of root-bearing axes attached to first order leafy shoots close to their attachment with the main leafy shoot. In each case, the root-bearing axes are smaller in diameter than the leafy shoots they are attached to and all root-bearing axes are aligned with the gravity vector. (**G-I**) Examples of thick sections showing the anisotomous branch point with attachment of a root-bearing axis and leafy shoot, the position of each thick section is illustrated on the reconstruction in (**D**), with arrow 1 (**G**), 2 (**H**), and 3 (**I**). (**J**) A bifurcating root-bearing axis with two apices attached to a larger leafy axis (leaves on large axis highlighted with arrowheads). (**K**) Higher magnification image of (**J**), showing the continuous cuticle covering the two apices and small leaf attached to the lower flank of the upper apex. 3D scale bar 1 × 0.1 × 0.1 cm (**D–F**), 2 × 1 × 1 mm (**A**). Scale bars, 5 mm (**G–I**), 2 mm (**B, C**) 1 mm (**J**), 500 µm (**K**). Specimen accession codes: Pb 4178 (**B**), Pb 4177 (**C**), Pb 4164 (**G**), Pb 4163 (**H**), Pb 4162 (**I**), Pb 2020_02 (**J, K**).

*1921*). The apex of a root-bearing axis had not been described previously. We searched for apices on axes with the characters of root-bearing axes and discovered an isolated bifurcating axis with two apices (*Figure 2J,K*). The apices were assigned to *A. mackiei* because of the presence of leaves on the parent axis (*Figure 2J*, arrowheads) and a small leaf on the flank of the upper apex (*Figure 2J, K*), *A. mackiei* is the only Rhynie chert plant with leaves. Both apices were covered by an unbroken cuticle, and a single small leaf was present on the upper apex, which together demonstrate that these are apices of root-bearing axes and not apices of rooting axes that lack leaves and cuticles (*Hetherington and Dolan, 2018a*), or leafy shoots where the apex is covered by a large number of leaves (*Edwards, 2003*; *Hueber, 1992*; *Kerp, 2018*; *Kerp et al., 2013*). These apices were found in a single thin section that was not part of a set of serial sections and therefore it was not possible to reconstruct the apex in 3D. The root-bearing axes described here are similar to the axes described as either root-bearing axes (*Matsunaga and Tomescu, 2016*; *Matsunaga and Tomescu, 2017*) or rhizomes (*Rayner, 1984*; *Schweitzer, 1980*; *Schweitzer and Giesen, 1980*; *Xu et al., 2013*) in other members of the Drepanophycales. The occurrence of these root-bearing axes in *A. mackiei* and

multiple other species highlights the conservation of body plans among members of the Drepanophycales.

## Rooting axes

*A. mackiei* rooting axes are similar to roots of extant lycopsids. However, they are designated rooting axes because they lack root hairs and their meristems lack a root cap and are consequently interpreted as transitional to the roots of extant plants (*Hetherington and Dolan, 2018a*; *Hetherington and Dolan, 2019*). These rooting axes include structures called rhizomes (*Bhutta, 1969*), small root-like rhizomes (*Kidston and Lang, 1920*; *Kidston and Lang, 1921*), and rooting axes (*Hetherington and Dolan, 2018a*; *Hetherington and Dolan, 2019*) of *A. mackiei* from previous descriptions of plant fragments. Rooting axes were always less than 2 mm in diameter and frequently less than 1 mm and were highly branched (*Bhutta, 1969*; *Hetherington and Dolan, 2018a*; *Kidston and Lang, 1920*; *Kidston and Lang, 1921*). Leaves, leaf traces, stomata, and cuticles were never found on rooting axes, even when the epidermis was well preserved (*Kidston and Lang, 1920*). The epidermis was frequently missing, suggesting that it was lost in older axes and the outer cortex was often limited to one or two cell layers (*Figure 1F*; *Kidston and Lang, 1920*). We found a single well-preserved highly branched rooting axis in our reconstruction

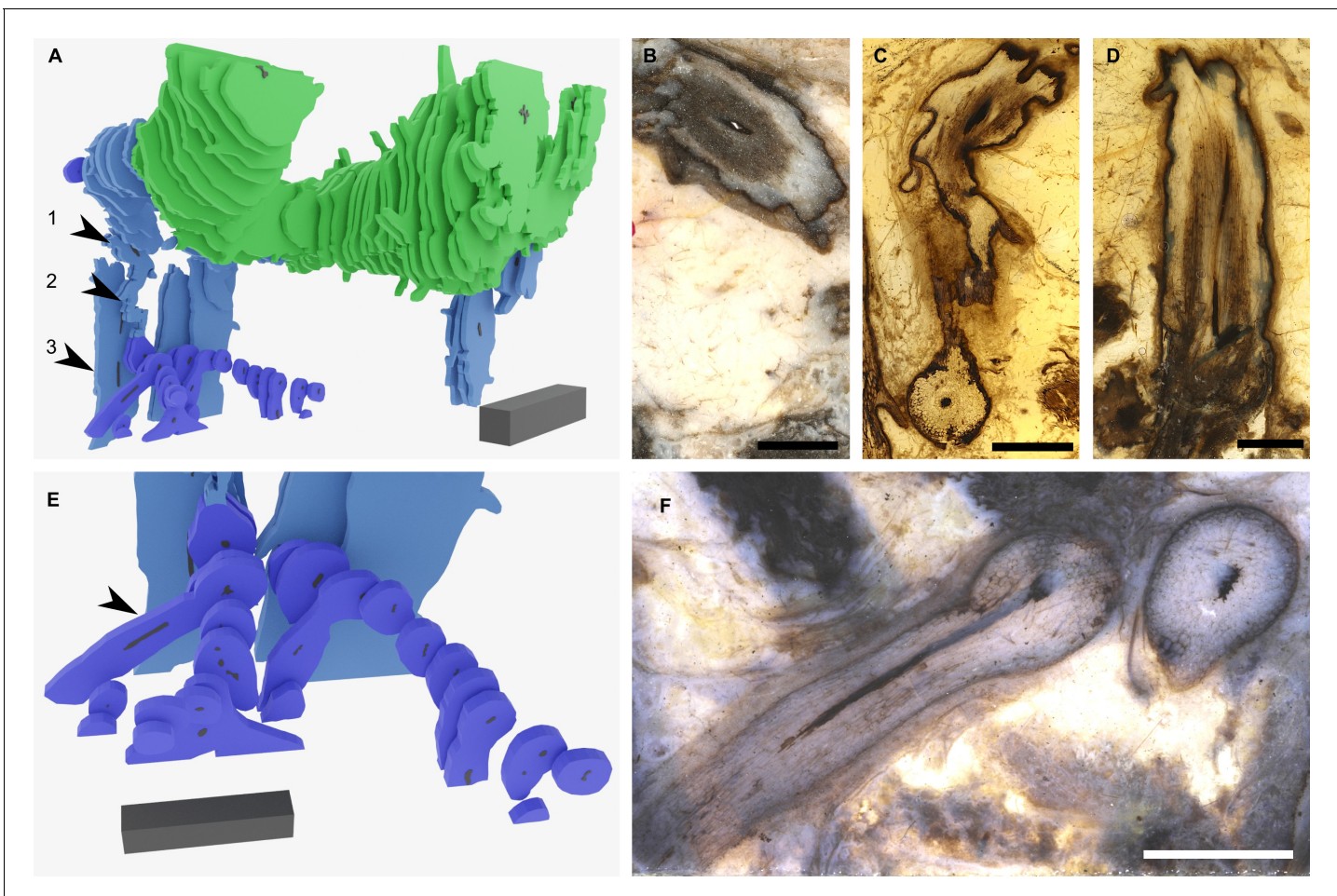

**Figure 3.** Rooting axes attached to root-bearing axes at anisotomous branch points. (A) Connection between a rooting axis in purple and root-bearing axis shown in blue based on the 3D reconstruction. (B–D) Three thick sections showing successive stages of the same root-bearing axis preserving the attachment of the rooting axis at an anisotomous branch point. The positions of the three thick sections in the reconstruction (A) are shown with the three numbered arrowheads 1 (B), 2 (C), and 3 (D). (E, F) Rooting axes branched profusely, through at least four orders of branching. (F) Example of branched rooting axis (marked by arrowhead in E). 3D scale bar 1 x 0.1 x 0.1 cm (A), 5 x 1 x 1 mm (E). Scale bars, 2 mm (B, C, D, F). Specimen accession codes: Pb 4174 (B, F), Pb 4175 (C), Pb 4177 (D).

(*Figure 3A–F*). This rooting axis was attached to a root-bearing axis at an anisotomous branch point (*Figure 3C*). There was a circular xylem strand at the centre of the rooting axis. The diameter of the rooting axis where it attached to the root-bearing axis was ca. 2 mm but decreased in size distally at successive branch points. Leaves, leaf traces, stomata, and cuticles were never observed on the rooting axis. The rooting axis was weakly gravitropic in contrast to the strong gravitropic growth observed in root-bearing axes (*Figure 3A,E*). The profuse branching of the rooting axis is evident with over four orders of branching preserved in less than 1 cm in our reconstruction (*Figure 3E*). We found no evidence that other axis types developed from the rooting axis. The morphological and anatomical boundary between the root-bearing axis and rooting axis was clear (*Figure 3C*); it involved the change from strong gravitropic growth to weak gravitropic growth, and the absence of scale leaves, stomata, and a well-marked cuticle all found on root-bearing axes but absent on rooting axes. This suggests that the boundary between the two axis types is defined at the point of branching and not in a continuum along a single axis. The rooting axis and its attachment to root-bearing axes described here correspond to the axes termed, roots (*Matsunaga and Tomescu, 2016*; *Matsunaga and Tomescu, 2017*; *Schweitzer, 1980*; *Schweitzer and Giesen, 1980*), root-like axes (*Gensel et al., 2001*; *Rayner, 1984*), and rootlets (*Xu et al., 2013*) in other members of the Drepanophycales. This suggests that the body plan of *A. mackiei* was similar to other members of the Drepanophycales.

Our new reconstruction from serial thick sections through an individual *A. mackiei* plant demonstrates that the *A. mackiei* body plan consisted of three distinct axis types – leafy shoot axes, root-bearing axes, and rooting axes – each with characteristic anatomy and morphology.

## Dichotomous origin of rooting axes

The rooting axes of *A. mackiei* hold a key position for interpreting the origin of roots in lycopsids because they were transitional between the ancestral rootless state and the derived state characterised by true roots with caps as found in extant lycopsids (*Hetherington and Dolan, 2018a*; *Hetherington and Dolan, 2019*). Our new reconstruction enables interpretation of these rooting axes and comparison with the rooting axes variously called roots, root-like axes, or rootlets in other members of the Drepanophycales that have been reported in the literature. Our reconstruction demonstrates that a number of characters are shared between the rooting axes of *A. mackiei* and rooting axes of other members of the Drepanophycales, including their attachment to root-bearing axes, weak gravitropic growth, and profuse dichotomous branching.

These shared characters suggest that the rooting system of *A. mackiei* was representative of the Drepanophycales and that inferences made with the exceptional preservation of *A. mackiei* can elucidate the characteristics of other members of the Drepanophycales that are preserved primarily as compression fossils. Our new reconstruction indicates that rooting axes connected to root-bearing axes at anisotomous branch points. Based on development of extant lycopsids (*Bierhorst, 1971*; *Fujinami et al., 2021*; *Gola, 2014*; *Guttenberg, 1966*; *Harrison et al., 2007*; *Hetherington and Dolan, 2017*; *Imaichi, 2008*; *Imaichi and Kato, 1989*; *Ogura, 1972*; *Ollgaard, 1979*; *Spencer et al., 2021*; *Yi and Kato, 2001*; *Yin and Meicenheimer, 2017*), there are two modes of branching that could produce anisotomous branch point morphology, endogenous branching, or dichotomous branching. Endogenous branching is the mode of branching where the meristem of the new axis develops from the internal tissues of the parent axis and breaks through the parent tissue to emerge, a mode of development typical of the initiation of roots of extant lycopsid species. Dichotomous branching is the mode of branching where the parent meristem splits in two to produce two daughter axes, a mode of development typical of roots, shoots, and rhizophores in extant lycopsids (*Bierhorst, 1971*; *Bruchmann, 1874*; *Gola, 2014*; *Guttenberg, 1966*; *Harrison et al., 2007*; *Hetherington and Dolan, 2017*; *Imaichi, 2008*; *Imaichi and Kato, 1989*; *Ogura, 1972*; *Ollgaard, 1979*; *Spencer et al., 2021*; *Wigglesworth, 1907*; *Yi and Kato, 2001*; *Yin and Meicenheimer, 2017*). To investigate which mode of development operated in *A. mackiei*, we examined anatomy of branch points.

If the rooting axes developed by endogenous branching from root-bearing axes, there would likely be a disruption to the tissues of the leafy shoot and evidence that the vascular trace of the root-bearing axes connected at right angles to the vascular trace of the leafy shoot (*Bruchmann, 1874*; *Guttenberg, 1966*; *Imaichi, 2008*; *Ogura, 1972*; *Van Tieghem and Douliot, 1888*; *Wigglesworth, 1907*; *Yi and Kato, 2001*). Our thick sections only provided a small organic

connection between the two axes (*Figure 3C*) limiting our ability to investigate the anatomical changes associated with branching. We therefore searched other Rhynie chert collections for rooting axes attached to root-bearing axes at anisotomous branching points. We reinvestigated an example originally described as a branching rhizome by *Bhutta, 1969*. The presence of scale leaves, a small number of leaf traces, and clear epidermis and cuticle on the main axis suggested that it was a root-bearing axis. Attached to this root-bearing axis was a smaller axis that we interpret as a rooting axis because of its rounded xylem, poorly preserved epidermis and the lack of both cuticle and root cap (*Figure 4*, *Figure 4—figure supplement 1*). We produced a 3D reconstruction of the anisotomous branch point that connects the two axes based on 119 peels (*Figure 4*, *Video 2*). Tissues were continuous between the root-bearing axis and rooting axis. The vascular trace for the rooting axis was seen to branch off and then run parallel to the main vascular trace before gradually arcing into the rooting axis (*Figure 4*). These characteristics, especially the dichotomy of the vascular trace, suggest that rooting axes developed from root-bearing axes by dichotomous branching.

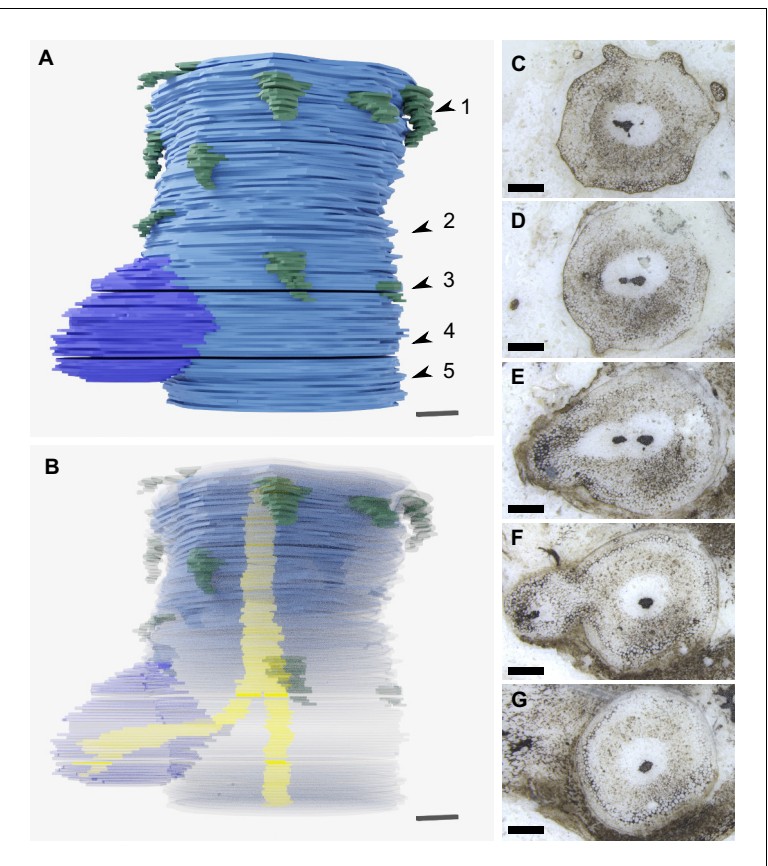

**Figure 4.** Rooting axes developed from root-bearing axes by dichotomous branching. (A) 3D reconstruction based on 119 peels from the A. Bhutta peel collection illustrating the attachment of a rooting axis to a root-bearing axis at an anisotomous branch point. Above the branch point the root-bearing axis, in blue, is covered by small-scale leaves indicated in dark green that are absent below the branch point. (B) Same 3D reconstruction as in (A) but with a transparent outline of the axis so the branching of the central xylem trace can be seen in yellow. (C–G) Images of representative peels used to create the 3D reconstruction showing the anatomical changes associated with branching, including the branching of the xylem strand (C, D), and the continuity of tissues between the root-bearing axis and the rooting axis (E, F). The positions of the peels (C–G) are shown with the numbered arrowheads 1–5 in (A). 3D scale bar 1 × 0.1 × 0.1 mm (A, B). Scale bars, 1 mm (C–G). A. Bhutta peel collection numbers RCA 14 (C), RCA 61 (D), RCA 81 (E), RCA 103 (F), RCA 114 (G).

The online version of this article includes the following figure supplement(s) for figure 4:

**Figure supplement 1.** Rooting axes lacked cuticles and root caps.

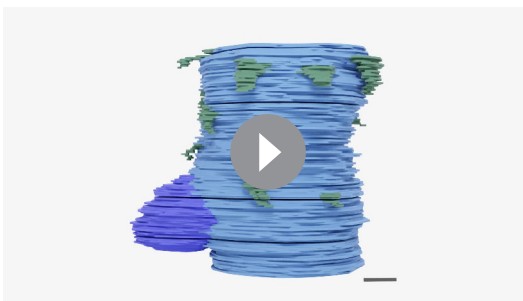

**Video 2.** 3D reconstruction of *A.mackiei* based on serial peels. A 3D reconstruction based on 119 peels from the A. Bhutta peel collection, University of Cardiff, illustrating the attachment of a rooting axis (purple) to a root-bearing axis (blue) at an anisotomous branch point. Small- scale leaves on the root-bearing axis indicated in dark green 3D scale bar 1 × 0.1 × 0.1 mm.
https://elifesciences.org/articles/69447#video2

While branching evidence in our reconstruction is consistent with the development of rooting axes from root-bearing axes by dichotomy, we tested if there was evidence for endogenous development because roots originate endogenously in extant lycopsids. Therefore, we searched for meristems of rooting axes preserved soon after they originated from root-bearing axes. We identified two fossilised *A. mackiei* meristems on a single thin section. This thin section preserves a large *A. mackiei* leafy shoot axis ca. 5 mm in diameter with stellate xylem cut in transverse section at the top of the image (*Figure 5A*, green arrowhead). Attached to the leafy shoot axis is a smaller root-bearing axis ca. 1.9 mm in diameter close to the attachment with the leafy shoot. This axis is identified as a root-bearing axis by the presence of a small scale leaf (*Figure 5A*, blue arrowhead, B), and its orientation aligned with the gravity vector based on a geopetally infilled void (*Figure 5—figure supplement 1*). Close to the base of the thin section are two apices on either side of the root-bearing axis (*Figure 5A*, white arrowheads). We interpret these as meristems because of their domed structure and the large number of small cells close to the apices. Given the attachment of these meristems to root-bearing axes and their small size, we interpret them as meristems of rooting axes. Cellular organisation of the promeristem (*Figure 5C–E*, *Figure 5—figure supplement 2*) is poorly preserved compared to other meristems we described (*Hetherington and Dolan, 2018a*), but the overall organisation including cell files running from the central vascular trace, and a cuticle covering the apices can be clearly recognised. There is no root cap as previously reported for rooting axes (*Hetherington and Dolan, 2018a*). Cell files are continuous between the root-bearing axes and rooting axes, and there is no evidence that the rooting axes initiated by endogenous branching and broke through the ground or dermal tissues of the root-bearing axes. The organisation of cells in these meristems is consistent with our hypothesis based on 3D reconstructed anatomy that the rooting axes developed by dichotomous branching from root-bearing axes.

## Discussion

We discovered that *A. mackiei* rooting axes originate by dichotomous branching. Furthermore, the branch connecting rooting axes and root-bearing axes was always anisotomous. This finding is significant because roots do not originate by dichotomy in extant lycopsids. Instead, roots of extant lycopsids originate endogenously from shoots, rhizophores, and rhizomorphs (*Bruchmann, 1874*; *Guttenberg, 1966*; *Hetherington and Dolan, 2017*; *Imaichi, 2008*; *Imaichi and Kato, 1989*; *Ogura, 1972*; *Yi and Kato, 2001*), and in rare cases by exogenous development in embryos, protocorms, or tubers where dichotomous branching does not occur (*Bower, 1908*; *Hetherington and Dolan, 2017*). The origin of *A. mackiei* rooting axes by anisotomous dichotomy was therefore different from the origin of roots in extant lycopsids, and this developmental mechanism is now extinct (*Figure 6*).

### Dichotomous branching giving rise to axes of different types is an ancient characteristic of lycophyte rooting systems

These reconstructions highlight that dichotomous branching, where the daughter axes produced at a branch point were of different types, was a characteristic of the rooting system of *A. mackiei*. Other Silurian and Devonian fossils suggest that this mode of branching may be a shared feature of many rooting systems in early-diverging lycophytes.

Nothia aphylla is an extinct early lycophyte closely related to the lycopsids (*Kenrick and Crane, 1997*; *Figure 6*). Its rooting system consisted of horizontal (plagiotropic) sporophyte axes with

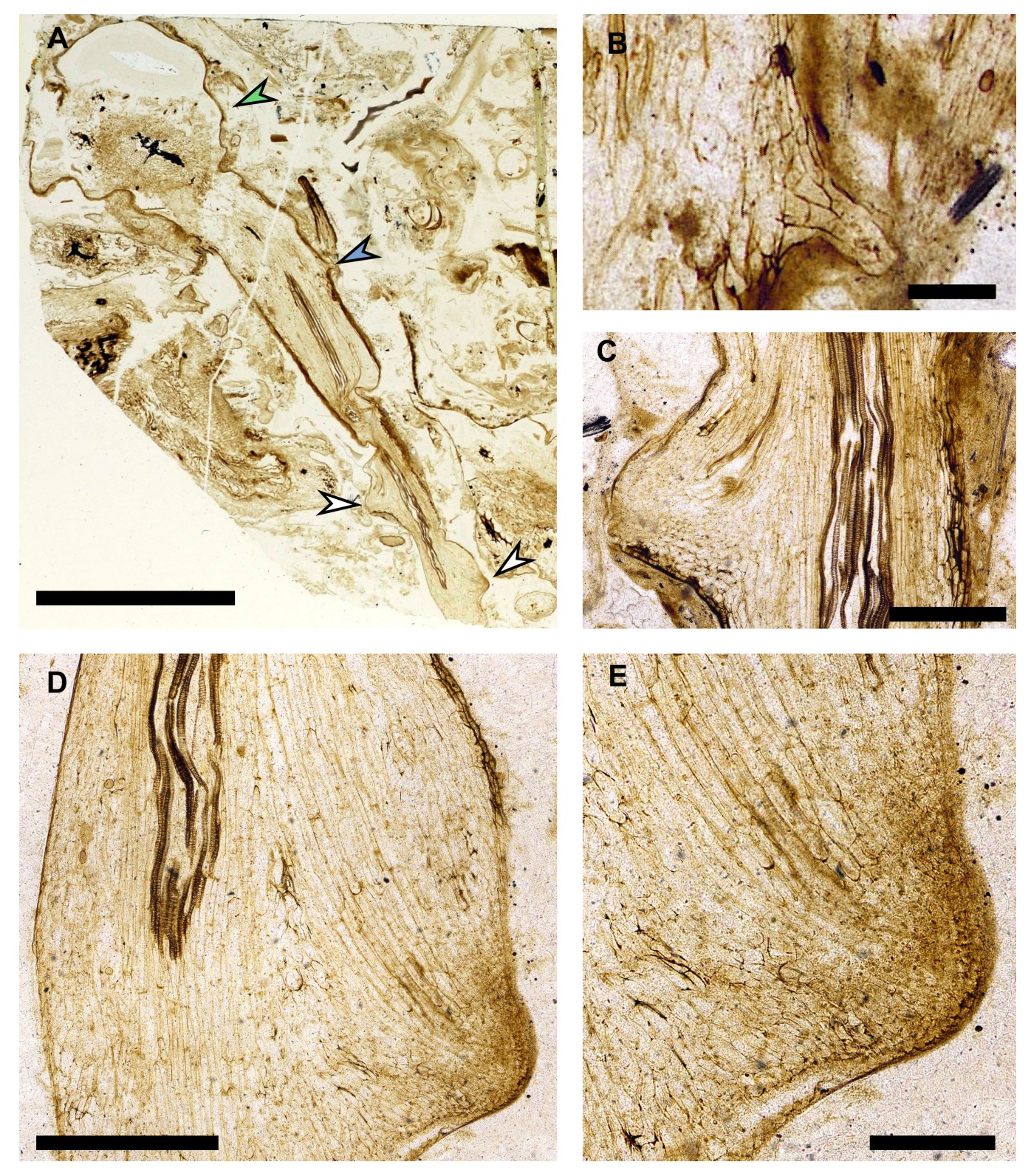

**Figure 5.** Fossilised meristems preserve evidence that rooting axes developed from root-bearing axes by anisotomous dichotomy. (A–E) *A. mackiei* axes preserving connection between a leafy shoot axis, root-bearing axis and two rooting axes apices. (A) Photograph of thin section NHMUK 16433 showing a large *A. mackiei* axis with stellate xylem cut in transverse section in the top left and highlighted with the green arrowhead, and attached root-bearing axis. On the root-bearing axis (A) the position of a scale leaf (B), is highlighted with a blue arrowhead and the two rooting axes meristems

*Figure 5 continued on next page*

*Figure 5 continued*

are highlighted with white arrowheads. (**C–E**) Rooting axis meristems marked by white arrowheads in (**A**) the left arrowhead (**C**) and right (**D, E**). (**D**) Well-preserved rooting axis meristem showing continuous cell files from the central vascular trace into the apex, the tissues of the rooting axis are continuous with the root-bearing axis indicating that development was by anisotomous dichotomy. Scale bars, 5 mm (**A**), 0.5 mm (**C, D**) 0.2 mm (**B, E**). Specimen accession codes: NHMUK 16433 (**A–E**).

The online version of this article includes the following figure supplement(s) for figure 5:

**Figure supplement 1.** A geopetally infilled void allowed growth orientation to be established.

**Figure supplement 2.** *A. mackiei* fossilised rooting axes meristems.

rhizoids, called rhizoidal sporophyte axes, that were both morphologically and anatomically different from the vertical (orthotropic) shoot axes (*Hetherington and Dolan, 2018b*; *Kerp et al., 2001*). The plagiotropic axes attach to the base of orthotropic axes at branch points (*Kerp et al., 2001*), suggesting that the transitions between orthotropic and plagiotropic growth occurred at branch points. This demonstrates that axis type could change at branch points in *N. aphylla* as it does in *A. mackiei*. The demonstration that this mode of branching was present outside of the lycopsids is consistent with it being an ancestral feature within the lycophytes.

Change in axis type at branch points was also a characteristic of the rooting systems of the zosterophylls, the other major grouping of extinct lycophytes. Zosterophylls are a paraphyletic or polyphyletic grouping of early-diverging lycophytes; however, most species group into a single clade sister to lycopsids (*Kenrick and Crane, 1997*; *Figure 6*). Many zosterophylls developed extensive rooting systems both morphologically and anatomically different from their shoot systems (*Edwards, 2003*; *Gensel et al., 2001*; *Hao et al., 2010*; *Hao et al., 2007*; *Kotyk and Basinger, 2000*; *Li, 1992*). In a number of fossils, rooting systems originated at branch points where one axis maintained a shooting character and the other acquired rooting characteristics. In the zosterophyll *Bathurstia denticulate* (*Gensel et al., 2001*; *Kotyk and Basinger, 2000*), rooting axes formed at k-branching points on shoot axes. This pattern of rooting axis development is also found in drepanophycalean lycopsids, and we show that it also occurs in the leafy shoot axes in *A. mackiei*. In a number of other *Zosterophyllum* species, axes are only identifiable as rooting axes because of changes in morphology or growth direction at a branch point where the two daughter axes are morphologically different from the original axis or grow in a different direction to the original axis; the rooting axis grows downwards, while the sister axis grows horizontally (*Gensel et al., 2001*; *Hao et al., 2007*; *Hao et al., 2010*). Finally, subordinate branching (*Kenrick and Crane, 1997*) in taxa such as *Gosslingia* (*Edwards, 1970*), *Tarella* (*Edwards and Kenrick, 1986*), *Deheubarthia* (*Edwards et al., 1989*), and *Hsüa* (*Li, 1992*) indicates that branching could result in the development of axes morphologically and anatomically different from the original axis, although not always associated with the origin of rooting systems. The frequent occurrence of rooting systems originating at branch points in the zosterophylls where axis type changed provides further support for it being an ancestral feature of all lycophytes. Testing this hypothesis is difficult due to the limited number of characteristics that can be used to distinguish between axes with a presumed rooting and shooting function in compression fossils (*Gensel et al., 2001*). However, it may be an ancestral character of lycophytes because this mode of branching is present in derived zosterophylls, such as *Bathurstia denticulate*, the drepanophycalean lycopsids and a number early-diverging lycophytes.

The roots of lycopsids evolved in a stepwise fashion, via transitional rooting axes similar to those preserved in *A. mackiei* (*Figure 6*) that originated at anisotomous branch points. The anisotomous origin of rooting axes is also likely to have occurred in other early-diverging lycophytes. However, this character does not occur in extant lycopsids. We therefore conclude that the origin of rooting axes through anisotomous dichotomy, a character of early-diverging lycophytes, has been lost during the evolution of lycopsids.

## Conclusions

We draw three significant conclusions from our 3D reconstruction of *A. mackiei*. (1) The body plan of *A. mackiei* was similar to the cosmopolitan members of the Drepanophycales found across America, Europe, and China in the Early and Middle Devonian (*Gensel et al., 2001*; *Li and Edwards, 1995*; *Matsunaga and Tomescu, 2016*; *Matsunaga and Tomescu, 2017*; *Rayner, 1984*; *Schweitzer, 1980*;

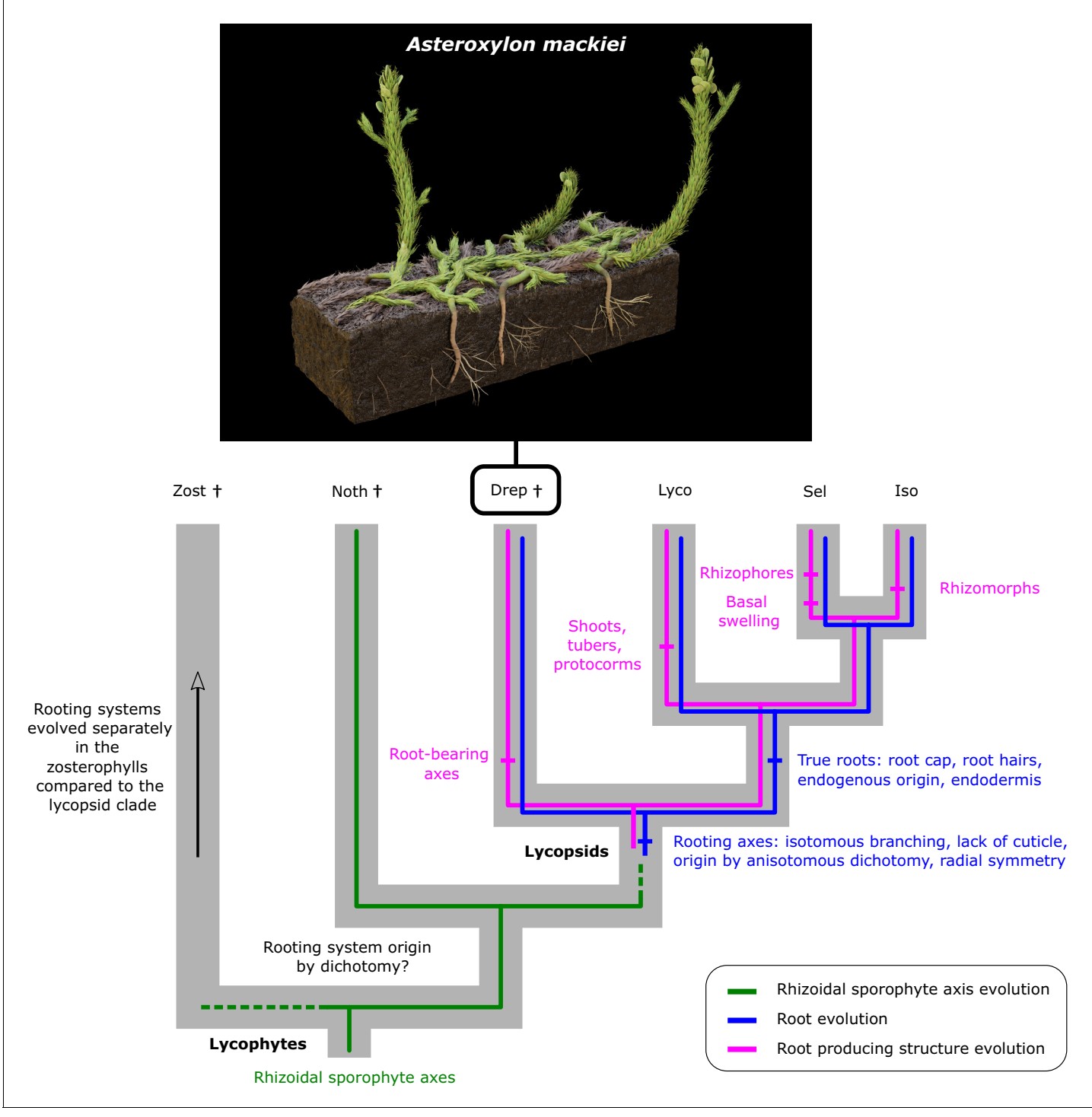

**Figure 6.** *A. mackiei* holds a key position for interpreting rooting system evolution within lycophytes. Cladogram of lycophytes with rooting system characteristics mapped on. Taxa abbreviations, Zost, zosterophylls, Noth *Nothia aphylla*, Drep, Drepanophycales, Lyco, Lycopodiales, Sel, Selaginellales, Iso, Isoetales. Extinct groups are indicated by dagger symbols (†). Cladogram after (*Kenrick and Crane, 1997*). *A. mackiei* illustration by Matt Humpage (https://twitter.com/Matt_Humpage). Leafy shoots in the illustration are roughly 1 cm in diameter.

*Schweitzer and Giesen, 1980*; *Xu et al., 2013*; *Xue et al., 2016*). This suggests that despite inhabiting the Rhynie geothermal wetland ecosystem (*Edwards et al., 2018*; *Garwood et al., 2020*; *Rice et al., 2002*; *Strullu-Derrien et al., 2019*; *Wellman, 2018*), mechanisms of body plan construction in *A. mackiei* likely also operated in other Devonian Drepanophycales. (2) We demonstrate that

rooting axes originated from root-bearing axes by dichotomy. In extant lycopsids, roots originate endogenously from shoots or specialised root producing structures, such as rhizophores (*Figure 6*). Once developed, roots, shoots, and rhizophores branch dichotomously (*Chomicki et al., 2017*; *Fujinami et al., 2021*; *Gola, 2014*; *Harrison et al., 2007*; *Hetherington and Dolan, 2017*; *Imaichi, 2008*; *Imaichi and Kato, 1991*; *Ollgaard, 1979*; *Yin and Meicenheimer, 2017*). However, the two daughter axes produced by dichotomous branching are always identical to the original axis: a shoot axis may branch dichotomously to form two identical shoot axes, a root axis may branch to form two identical root axes, and a rhizophore branches to form two identical rhizophores (*Chomicki et al., 2017*; *Fujinami et al., 2021*; *Gola, 2014*; *Harrison et al., 2007*; *Hetherington and Dolan, 2017*; *Imaichi, 2008*; *Imaichi and Kato, 1991*; *Ollgaard, 1979*; *Yin and Meicenheimer, 2017*). In *A. mackiei,* the root-bearing axes branched anisotomously to produce one root-bearing axis and a rooting axis. Our findings therefore suggest that anisotomous dichotomy was key for the development of the complex body plan of *A. mackiei*, which builds on previous suggestions that the evolution of anisotomous dichotomy in land plants was a key developmental innovation for both the evolution of leaves (*Sanders et al., 2007*; *Stewart and Rothwell, 1993*; *Zimmerman, 1952*) and rooting systems (*Gensel et al., 2001*). (3) Finally, these findings demonstrate how 3D evidenced-based reconstructions of the Rhynie chert plants can define how these plants grew and developed. These reconstructions allow the body plans of Rhynie chert plants to be compared with plants preserved as compression fossils where body plans can be determined, but cellular anatomy is not preserved.

Taken together, our 3D reconstruction demonstrates that the body plan of *A. mackiei* comprised three distinct axis types, and we demonstrate that roots developed through anisotomous dichotomy of a specialised root-bearing axis. This mode of rooting system development is now extinct, but played a key role in the development of the complex rooting systems of the Drepanophycales.

## Materials and methods

Specimen accession code abbreviations: Forschungsstelle für Paläobotanik, Institut für Geologie und Paläontologie, Westfälische Wilhelms-Universität, Münster, Germany; Pb. The Hunterian, University of Glasgow, GLAHM. Natural History Museum, London, NHMUK.

### Thick section preparation

The reconstruction of *A. mackiei* was based on a series of 31 thick sections made from a single block of chert collected from a trench dug in 1964. Thick sections were made by mounting the rock to 2.8 cm by 4.8 cm glass slides using thermoplastic synthetic resin and cutting with a 1 mm thick diamond blade. The resulting thin wafer of rock was ground with silicon carbide powder to ensure a flat surface, a number of specimens were released from the glass slide and turned around to grind them down further from the other side (*Hass and Rowe, 1999*). Thick sections were not sealed with a cover slip and were deposited in the collection of the Forschungsstelle für Geologie und Paläontologie, Westfälische Wilhelms-Universität, Münster, Germany, under the accession numbers Pb 4161–4191.

### 3D reconstruction of *A. mackiei* from thick sections

To create a 3D reconstruction based on the series of thick sections, photographs of the upper and lower surface of the thick sections were taken. Thick sections were placed on a milk glass pane above a lightbox and incident light was provided by two lamps (*Kerp and Bomfleur, 2011*). The surface of the specimen was covered with cedar wood oil and images were captured with a Canon MP-E 65 mm macro lens and a Canon EOS 5D Mark IV single-lens reflex camera. Images of the full series of thick sections were deposited on Zenodo (http://doi.org/10.5281/zenodo.4287297). Line drawings were made of both the outline of the *A. mackiei* axes of interest and also the central vascular trace in each axis using Inkscape (https://inkscape.org/). Line drawings were imported into Blender (https://www.blender.org/) and extruded in the z-dimension by 0.2 mm to turn each outline into a 3D slice. In the model, a thick section was then represented by an upper and lower slice of 0.2 mm separated by a gap of 0.4 mm. Slices from consecutive thick sections were aligned and a 1 mm gap was left to represent the material lost to the saw blade when making the sections. Images and

animations of the reconstruction were made in Blender, and the 3D reconstruction was deposited on Zenodo (http://doi.org/10.5281/zenodo.4287297).

### 3D reconstruction of *A. mackiei* from peels

A branching root-bearing axis was reconstructed from a series of RCA 1–119 from the A. Bhutta collection (*Bhutta, 1969*) at the University of Cardiff. Four peels were missing from the series, RCA 3, RCA 31, RCA 80, and RCA 102. Images of peels were scanned using an Epson perfection V500. Images of the full series of peels were deposited on Zenodo (http://doi.org/10.5281/zenodo. 4287297). Line drawings were made of the outline of the *A. mackiei* axis of interest and also the central vascular trace in each axis using Inkscape (https://inkscape.org/). Line drawings were imported into blender and extruded in the z-dimension by 0.058 mm based on *Bhutta, 1969*. Consecutive slices were aligned to produce the 3D model, and gaps were left for the four missing peels. The 3D reconstruction was deposited on Zenodo (http://doi.org/10.5281/zenodo.4287297).

### Higher magnification images and microscopy

Thick sections were placed on a milk glass pane above a lightbox and incident light was provided by two lamps and the surface covered in cedar wood oil (*Kerp and Bomfleur, 2011*). Photographs were taken with a Canon EOS 5D Mark IV digital single-lens reflex camera mounted on a copy stand using either a Canon MP-E 65 mm or Canon EFS 60 mm macro lens (*Figure 1H–J*, *Figure 2B,C,G– K*, *Figure 3B–D,F*, *Figure 1—figure supplement 1A,B*, *Figure 1—figure supplement 2*). The photograph of thin section NHMUK 16433 (*Figure 5C*, *Figure 5—figure supplement 1A*) was taken with a Nikon D80 camera with a 60 mm macro lens mounted on a copystand with light from below from a lightbox. Higher magnification images were taken of the branching *A. mackiei* axis from the A. Bhutta collection with a Leica M165 FC with light from above provided by a Leica LED ring illuminator. (*Figure 4C–G*, *Figure 4—figure supplement 1*). Microscope images of NHMUK 16433 (*Figure 5D–G*, *Figure 5—figure supplement 1B*, *Figure 5—figure supplement 2*) were taken with a Nikon Eclipse LV100ND.

## Acknowledgements

We would like to thank D Edwards for access to the Bhutta collection at the University of Cardiff, P Hayes for access to collection at the NMHUK, M Humpage for illustrations in *Figure 1G* and the illustration of *A. mackiei* in *Figure 6*, and three reviewers for their constructive comments.

## Additional information

### Funding

| Funder | Grant reference number | Author |
|---|---|---|
| Magdalen College, University of Oxford | George Grosvenor Freeman Fellowship by Examination in Sciences | Alexander J Hetherington |
| University of Oxford | Hester Cordelia Parsons Fund | Alexander J Hetherington |
| UK Research and Innovation | MR/T018585/1 | Alexander J Hetherington |
| Biotechnology and Biological Sciences Research Council | Research Experience Placement | Anna Lee Jones |
| European Commission | 250284 | Liam Dolan |
| European Commission | 787613 | Liam Dolan |
| European Commission | 238640 | Hans Kerp Liam Dolan |
| Deutsche Forschungsgemeinschaft | KE 584/13- 1 | Hagen Hass Hans Kerp |
| Deutsche Forschungsgemeinschaft | KE 584/13- 2 | Hagen Hass Hans Kerp |

The funders had no role in study design, data collection and interpretation, or the decision to submit the work for publication.

## Author contributions
Alexander J Hetherington, Conceptualization, Formal analysis, Supervision, Funding acquisition, Investigation, Methodology, Writing - original draft, Writing - review and editing; Siobhán L Bridson, Anna Lee Jones, Formal analysis, Investigation, Methodology, Writing - review and editing; Hagen Hass, Resources, Methodology; Hans Kerp, Resources, Data curation, Funding acquisition, Methodology, Writing - review and editing; Liam Dolan, Conceptualization, Supervision, Funding acquisition, Writing - original draft, Writing - review and editing

## Author ORCIDs
Alexander J Hetherington (iD) https://orcid.org/0000-0002-1687-818X
Liam Dolan (iD) https://orcid.org/0000-0003-1206-7096

## Decision letter and Author response
Decision letter https://doi.org/10.7554/eLife.69447.sa1
Author response https://doi.org/10.7554/eLife.69447.sa2

## Additional files
### Supplementary files
• Transparent reporting form

## Data availability
Fossil preparations described in this study are housed in the A. Bhutta collection at the University of Cardiff, UK. Forschungsstelle für Paläobotanik, Institut für Geologie und Paläontologie, Westfälische Wilhelms-Universität, Münster, Germany. The Hunterian, University of Glasgow, UK and the Natural History Museum, London, UK. Photographs of the series of thick sections and peels used to create 3D reconstructions of A. mackiei have been deposited on Zenodo (http://doi.org/10.5281/zenodo.4287297). All other data supporting the findings of this study are included in the paper.

The following dataset was generated:

| Author(s) | Year | Dataset title | Dataset URL | Database and Identifier |
|---|---|---|---|---|
| Hetherington AJ, Bridson SL, Jones AL, Hass H, Kerp H, Dolan L | 2020 | Supplementary information: An evidence-based 3D reconstruction of Asteroxylon mackiei the most complex plant preserved from the Rhynie chert | https://zenodo.org/record/4287297#.YIPNQ5BKhPY | Zenodo, 10.5281/zenodo.4287297 |

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
