## [Decision Letter]

**Acceptance summary:**

Using novel 3D reconstruction techniques, the authors provide the first evidence-based reconstruction of *Asteroxylon mackiei*, the Early Devonian Rhynie chert plant known for a century, and demonstrate that it possessed an extinct pattern of root developmental transitional to the evolution of true roots in modern club-mosses. The use of multiple lines of evidence and 3D reconstructions based on serial sections of petrified materials provides compelling support for the key conclusions of the paper. This paper will be of interest to readers in the field of plant evolutionary biology and paleontology, paleobotany in particular.

**Decision letter after peer review:**

Thank you for submitting your article "An evidence-based 3D reconstruction of *Asteroxylon mackiei* the most complex plant preserved from the Rhynie chert" for consideration by *eLife*. Your article has been reviewed by 3 peer reviewers, including Min Zhu as the Reviewing Editor and Reviewer #1, and the evaluation has been overseen by Jürgen Kleine-Vehn as the Senior Editor. The following individuals involved in review of your submission have agreed to reveal their identity: Carol Hotton (Reviewer #2); Paul Kenrick (Reviewer #3).

Essential revisions:

1) The discussion part of this manuscript should be strengthened to a larger extent. We suggest to add a figure in which characters discussed in text are plotted onto a simple outline phylogenetic diagram of lycopods, living and extinct. A figure of this sort would also give the work greater impact.

2) Parallelism, such as a potential parallel between the development of the root bearing axis of Asteroxylon and a type of branch development that is widely seen in zosterophylls, is not well addressed in the present version.

*Reviewer #3 (Recommendations for the authors):*

The work is very well written and presented, so I don't have very many suggestions for how the science and presentation might be strengthened. I make two suggestions below.

1. The authors state in the abstract that this is '…the first complete reconstruction of the lycopsid *Asteroxylon mackiei*…'. I don't think that their work bears this statement out. It would be more accurate to say that it is the first well-evidenced reconstruction of the structure and development of the rooting system. That is the focus of the paper, and it is sufficient on its own. The evidence of the structure of the aerial system, as presented in Figure 1G, is not as complete or as compelling.

2. I was struck by a potential parallel between the development of the root bearing axis of Asteroxylon and a type of branch development that is widely seen in zosterophylls, which are the extinct sister group of lycopods. See, for example, the discussion of subordinate branching on pages 152-157 of Kenrick and Crane (1997). Some of the lateral branches in zosterophylls appear to be undeveloped aerial axes, but in plants like Tarella and Sawdonia some might have developed into rooting structures. The authors could consider adding a few sentences on this.

Kenrick P, Crane PR. 1997. The origin and early diversification of land plants: a cladistic study. Washington: Smithsonian Institution Press.

---

## [Author Response]

Essential revisions:1) The discussion part of this manuscript should be strengthened to a larger extent. We suggest to add a figure in which characters discussed in text are plotted onto a simple outline phylogenetic diagram of lycopods, living and extinct. A figure of this sort would also give the work greater impact.

We have extended and added an additional figure to the discussion.

2) Parallelism, such as a potential parallel between the development of the root bearing axis of Asteroxylon and a type of branch development that is widely seen in zosterophylls, is not well addressed in the present version.

This has been added to the discussion.

Reviewer #3 (Recommendations for the authors):The work is very well written and presented, so I don't have very many suggestions for how the science and presentation might be strengthened. I make two suggestions below.1. The authors state in the abstract that this is '…the first complete reconstruction of the lycopsid Asteroxylon mackiei…'. I don't think that their work bears this statement out. It would be more accurate to say that it is the first well-evidenced reconstruction of the structure and development of the rooting system. That is the focus of the paper, and it is sufficient on its own. The evidence of the structure of the aerial system, as presented in Figure 1G, is not as complete or as compelling.

We have changed this in the abstract, and updated Figure 1G accordingly. A reconstruction of the aerial system of Asteroxylon is now only given in the discussion rather than the results.

2. I was struck by a potential parallel between the development of the root bearing axis of Asteroxylon and a type of branch development that is widely seen in zosterophylls, which are the extinct sister group of lycopods. See, for example, the discussion of subordinate branching on pages 152-157 of Kenrick and Crane (1997). Some of the lateral branches in zosterophylls appear to be undeveloped aerial axes, but in plants like Tarella and Sawdonia some might have developed into rooting structures. The authors could consider adding a few sentences on this.Kenrick P, Crane PR. 1997. The origin and early diversification of land plants: a cladistic study. Washington: Smithsonian Institution Press.

This has been added to the discussion of the paper.